# Effect of Intrahippocampal Administration of α7 Subtype Nicotinic Receptor Agonist PNU-282987 and Its Solvent Dimethyl Sulfoxide on the Efficiency of Hypoxic Preconditioning in Rats

**DOI:** 10.3390/molecules26237387

**Published:** 2021-12-06

**Authors:** Elena I. Zakharova, Andrey T. Proshin, Mikhail Y. Monakov, Alexander M. Dudchenko

**Affiliations:** 1Laboratory of General Pathology of Cardiorespiratory System, Institute of General Pathology and Pathophysiology, Baltiyskaya, 8, 125315 Moscow, Russia; monakovm@mail.ru (M.Y.M.); amdudchenko@gmail.com (A.M.D.); 2Laboratory of Functional Neurochemistry, P.K. Anokhin Institute of Normal Physiology, Baltiyskaya, 8, 125315 Moscow, Russia; proshin_at@mail.ru

**Keywords:** hypoxic preconditioning, hippocampus, CA1 area, intrahippocampal injections, nicotinic α7 receptors (α7nAChRs), PNU-282987 (PNU), dimethyl sulfoxide (DMSO)

## Abstract

We have previously suggested a key role of the hippocampus in the preconditioning action of moderate hypobaric hypoxia (HBH). The preconditioning efficiency of HBH is associated with acoustic startle prepulse inhibition (PPI). In rats with PPI > 40%, HBH activates the cholinergic projections of hippocampus, and PNU-282987, a selective agonist of α7 nicotinic receptors (α7nAChRs), reduces the HBH efficiency and potentiating effect on HBH of its solvent dimethyl sulfoxide (DMSO, anticholinesterase agent) when administered intraperitoneally. In order to validate the hippocampus as a key structure in the mechanism of hypoxic preconditioning and research a significance of α7nAChR activation in the hypoxic preconditioning, we performed an in vivo pharmacological study of intrahippocampal injections of PNU-282987 into the CA1 area on HBH efficiency in rats with PPI ≥ 40%. We found that PNU-282987 (30 μM) reduced HBH efficiency as with intraperitoneal administration, while DMSO (0.05%) still potentiated this effect. Thus, direct evidence of the key role of the hippocampus in the preconditioning effect of HBH and some details of this mechanism were obtained in rats with PPI ≥ 40%. The activation of α7nAChRs is not involved in the cholinergic signaling initiated by HBH or DMSO via any route of administration. Possible ways of the potentiating action of DMSO on HBH efficiency and its dependence on α7nAChRs are discussed.

## 1. Introduction

Nicotinic acetylcholine receptors (nAChRs), the diversity of which was discovered at the end of the 20th century [1], are attracting increasing research attention due to their active participation in a number of brain functions (primarily cognitive), in health and pathogenesis [2,3,4,5,6,7,8]. The α7 subtype of nAChRs is of the greatest interest, especially hippocampal α7nAChRs [2,3,4,5,6,9]. It is known that the hippocampus is one of the central brain structures in the processes of learning and memory consolidation, and α7 is the most common receptor subtype in this structure [10,11,12].

The involvement of α7nAChRs in the pathogenesis of mental and neurological diseases makes them an important therapeutic target. In both ex vivo and in vivo models of cerebral ischemia, pre-activation of α7nAChRs has a protective effect on neuronal ischaemic damage [13,14]. Another of the main methods of anti-ischaemic and antihypoxic protection is hypoxic/ischaemic preconditioning, an exposure to moderate hypoxic/ischaemic stimuli that increases resistance to severe hypoxic/ischaemic shock [15,16,17]. When using specific adaptive stimuli, it was found that the preconditioning action acts via α7nAChRs [18,19]. This suggests the involvement of α7nAChRs in the implementation of hypoxic preconditioning.

In order to investigate the neuronal mechanisms of hypoxic preconditioning, we used the model of a one-time moderate hypobaric hypoxia (HBH) in rats. The hypoxic preconditioning effect of HBH is a short-term adaptation to severe hypoxic impacts and its adaptive effect persists for twenty-four hours [20,21]. We have previously shown that the efficiency of HBH, measured by the time (T) of resistance to severe hypobaric hypoxia (SHBH, 4.5% O_2_), is associated with the level of acoustic startle prepulse inhibition (PPI) [22,23], “the PPI-T test”. Using this test, we identified some peculiarities of cholinergic organization in the preconditioning mechanisms of HBH, which are masked when analyzing the experimental data as a whole.

In our pharmacological experiments, we investigated the effects of selective α7nAChR agonist PNU-282987 (PNU) [24] on the efficiency of HBH using a single intraperitoneal (IP) injection before HBH [23,25]. PNU showed a dose-dependent PPI-associated effect on the T values, the direction of which changed at a threshold PPI of approximately 40%. dimethyl sulfoxide (DMSO), a bipolar aprotic solvent for hydrophobic drugs, including PNU, at an initial concentration of 2–3%, showed the PPI-associated effects on the T values at the same threshold PPI. DMSO potentiated the effects of HBH in rats with higher PPI values (>40%) and reduced it in rats with lower PPI values (<40%). PNU, in contrast, reduced the effects of DMSO on HBH in both rat subgroups and, at high doses of 1,2–12 µmol/kg, reduced the effects of HBH in rats with PPI > 40% and potentiated the effects in rats with PPI < 40%.

The principles for this bi-directionality of both substances on the efficiency of HBH were not well understood. Therefore, based on a stable correlation between the HBH efficiency and PPI values, we conducted a biochemical study of the central cholinergic mechanisms of HBH in rats using PPI-T test readings [25]. In response to HBH, a cholinergic reaction associated with PPI was detected in the synaptosomes of the caudal brain (medulla oblongata and pons Varolii), neocortex, and hippocampus. However, a bi-directional cholinergic response to HBH was manifested only in the hippocampus, similar to that observed in pharmacological experiments. Selectively in this brain structure, activation of membrane-bound choline acetyltransferase (a marker of cholinergic neurons and their transmitter function) in rats with PPI > 40% and inhibition of enzyme activity in rats with PPI < 40% was observed in the synapses of projection neurons from forebrain nuclei (fraction of “light” synaptosomes). These data indicated the key role of the hippocampus and its cholinergic projection system in the mechanism of hypoxic preconditioning of HBH.

Thus, our studies revealed that the increase in resistance to oxygen deprivation during preconditioning is achieved by two opposing cholinergic mechanisms, depending on the PPI value of a subject. Comparison of biochemical and pharmacological data suggested that DMSO, at final concentrations in the brain of no more than 0.04%, has an anticholinesterase effect [26,27], and that PNU effects are probably due to desensitization of α7nAChRs, which is characteristic of nAChR agonists at a wide range of concentrations [28]. This could explain the physiological response to PNU, which is opposite to the action of HBH on the cholinergic system of the hippocampus and the potentiating effect of DMSO on HBH efficiency.

In order to validate the hippocampus as a key structure in the mechanism of hypoxic preconditioning and reveal a significance of activation of α7nAChRs, we performed an in vivo study involving the intrahippocampal (iHippo) injection of PNU and its solvent DMSO into the CA1 area. We believed that PNU at a concentration of 30 μM in the hippocampus should manifest an activating effect on interneurons of the hippocampus, as demonstrated in vitro in cultured hippocampal neurons [3], and we would be able to assess the physiological effect of α7nAChRs activation under hypoxic preconditioning. The study was performed on a group of rats with PPI ≥ 40%. This is the case in most mentally healthy animals and humans [29,30]. In addition to this, the relatively low efficiency of HBH found in rats with PPI > 40% makes it especially urgent to identify preconditioning mechanisms and search for means that could potentiate them.

## 2. Results

### 2.1. Postmortem Brain Analysis of Rats Given iHippo Injections

Visual analysis of brain sections demonstrated the correctness of the insertion sites of the guide cannulas to the selected coordinates (Figure 1). The injection sites were in the CA1 area of the hippocampus.

### 2.2. Pharmacological Data Analysis

#### 2.2.1. Control Group

In the saline IP and iHippo groups, the T values obtained in this study in rats with PPI ≥ 40% (n = 7 in the saline IP group, n = 3 in the saline iHippo group) as well as an one T value as a “pure control” for surgical procedures, showed similar levels of variation. These data were fully integrated into the previously obtained dataset on the PPI-T test in rats with PPI ≥ 40% (n = 17) [25] (Figure 2). 

Expressed as a percentage of the trend line, these data also showed similar means ± SEM% (Figure 3). Therefore, we pooled the T values of both the saline and “pure control” groups obtained in this study and used it as a control group for comparison with the PNU and DMSO groups (n = 11, means ± SEM% = 101.5 ± 10.4%; means ± SEM = 726.1 ± 71.3 s). Note that when comparing with the array of all control data in rats with PPI ≥ 40% (n = 28), we obtained the same comparative statistical results.

#### 2.2.2. DMSO Groups

Since the α7nAChR agonist always acts against the background of DMSO, it is advisable to present the results in comparison to the solvent (Figure 4). Both DMSO groups included predominantly animals with relatively high PPI values ≥ 70%, data for which were absent in previous studies. Therefore, although the totality of all T values represents a single variation series, we did not include the DMSO values obtained earlier in the IP group.

About 3% DMSO in the IP group (n = 6) increased the HBH efficiency by 79% with the same significance as in previous studies in rats with PPI > 40% (*p* < 0.05) [23,25]. In the iHippo group (0.05%, n = 8), DMSO showed a slightly more potent potentiating effect on HBH efficiency (by 111%) (Figure 4). The difference between the two DMSO groups was not significant. However, the difference between the iHippo group and the control group was more significant (*p* < 0.02). 

The DMSO efficiency vector shows an inverse relationship with the HBH efficiency vector, namely: the higher the T values in both DMSO groups, the lower the HBH efficiency (and the higher the PPI values [23,25]). Therefore, we performed a correlation analysis between the DMSO and control groups. To do this, in conjunction with each T value in the DMSO groups, we matched the T value in the control group with the PPI value ±5–8% (Table 1). The comparison revealed a negative correlation between the control group and the combined DMSO group, as well as with each DMSO group separately. 

#### 2.2.3. PNU iHippo Group

In this study, the action of the α7nAChR agonist PNU was assessed only with iHippo administration (30 µM, n = 6). Therefore, the effects of PNU were compared with the DMSO iHippo group. PNU significantly eliminated the effect of DMSO on HBH efficiency (Figure 5). 

The average T values in the iHippo PNU group were 16% higher than those for the IP PNU group in previous experiments at high concentrations of 300–3000 nmol/mL (1.2–12 µmol/kg) [25], but this difference was not significant. Means ± SEM in the PNU groups at equal levels of PPI were 74.8 ± 5.0% as a percentage of the control in the iHippo PNU group (n = 5) versus 64.6 ± 11.3% in IP PNU group (n = 11). Overall, the intergroup relationships in the control-DMSO-PNU groups were very similar between the iHippo and IP routes of administration (Figure 5, [25]). 

Correlation analysis of T values in the iHippo PNU group did not reveal significant correlations with either the control group or the iHippo DMSO group.

## 3. Discussion

In a previous biochemical study, we obtained results indicating a key role of the hippocampus and its cholinergic projection system in the formation of hypoxic preconditioning in rats in subgroups with PPI both greater and lower than 40% [25]. The efficiency of HBH was influenced by the selective α7nAChR agonist PNU and its solvent DMSO upon IP administration in both subgroups of rats [23,25]. In order to validate the key role of the hippocampus in the preconditioning mechanisms of HBH and investigate the cholinergic component of this mechanism, primarily the role of α7nAChR activation, we performed a study on the effect of iHippo administration of PNU and DMSO on the efficiency of HBH preconditioning in rats with PPI ≥ 40%.

### 3.1. The Hippocampus Is A Key Structure in the Mechanism of the Preconditioning Action of HBH

iHippo introduction of the substances made it possible to obtain these data. Both the α7nAChR agonist and its solvent DMSO had the same effect on the efficacy of HBH as when administered IP.

This, together with our previous biochemical study [25], represents the first evidence not only of the involvement of the hippocampus in the mechanisms of hypoxic preconditioning, but also of its key role in this process.

At the same time, the interaction between PNU and DMSO in the preconditioning mechanism of HBH was shown to be more complicated than we had previously assumed [23,25].

### 3.2. Comparison of PNU Effects upon iHippo and IP Administration; α7nAChR Activation Is Not Involved in the Cholinergic Activation of Hypoxic Preconditioning 

Selective α7nAChR agonist PNU exhibited the same effect on iHippo administration as with IP administration, acting to oppose the action of DMSO and HBH [23,25]. According to the literature, 30 μM PNU is the concentration at which the agonist induces maximum cell currents in cultured rat hippocampal neurons sensitive to the selective α7nAChR antagonist methyllycaconitine [3,24]. Thus, we believe that, in our study, when PNU was iHippo administered, we observed a consequence of α7nAChRs activation. 

Given the similarity of the effects of PNU upon iHippo and IP administration, we conclude that the α7nAChR activation triggered PNU effects in the hippocampus as well upon IP administration in the concentration range of 26–3000 nmol/mL (104 nmol–12 µmol/kg). Thus, our assumption about the desensitizing effect of PNU under HBH conditions was erroneous [23,25]. This seemed logical to us, since (1) activation of cholinergic projection neurons in the hippocampus in rats with PPI > 40% and, conversely, inhibition of their activity in rats with PPI < 40% was detected; (2) this coincided with the bidirectional effects of PNU; and (3) PNU at high concentrations suppressed the efficiency of HBH in rats with PPI > 40% and potentiated it in rats with PPI < 40% [25].

Thus, cholinergic activation via α7nAChRs interferes with HBH-initiated cholinergic signaling in rats with PPI ≥ 40%. This paradox can be explained only by the fact that α7nAChRs are not involved in the preconditioning mechanism of cholinergic activation in the hippocampus. This completely changes our understanding of the possible role of α7nAChRs in the mechanism of hypoxic preconditioning, and we propose a different explanation for the obtained data.

### 3.3. Under HBH Conditions, α7nAChRs May Be Involved in the Mechanisms of Filtering Cognitive Stimuli

The receptor organization of brain functions is an integral part of their neuronal organization. In the hippocampus, interneurons predominantly express α7nAChRs [10,28]. In electrophysiological studies in vitro, α7nAChRs in the CA1 zone activate inhibitory GABAergic interneurons and thereby indirectly inhibit the activity of glutamatergic pyramidal neurons. This is a typical reaction of pyramidal neurons in the hippocampus in response to the activation of α7nAChRs recorded in CA1 [3,28,31,32,33]. The direct participation of this mechanism in the initiation of early and late long-term potentiation (LTP) has been demonstrated, and the activation of α7nAChRs in this neuronal chain is a necessary link [5,8,32,33].

The participation of the hippocampus in the adaptive plasticity of visceral functions has previously been demonstrated [34]. Neuronal networks of the brain are the most complex and least understood problem in the mechanisms of nerve functions. Nevertheless, there is no doubt that HBH in the hippocampus is realized by neural pathways and/or interactions, including cholinergic ones, separate from the pathways of memory consolidation in cognitive functions. We discussed in detail in our previous article that, according to modern concepts, in order to perceive a sufficiently strong external signal capable of initiating nervous function, there is a system of “filtering” signals in the brain and the hippocampus is responsible for filtering and maintaining new and significant signals of any modality [25]. The essence of filtering is inhibition of the neuronal response to stimuli that are insignificant in order to process the dominant stimulus [35,36,37]. 

According to our data, in rats with PPI > 40%, activation of α7nAChRs prevents the implementation of hypoxic preconditioning. Therefore, it is possible that the natural response to HBH in the hippocampus of these animals is to suppress the action of α7nAChRs. Using another experimental model, we obtained data that can be explained by the mechanism of filtering cognitive stimuli through α7nAChRs in conditions of brain oxygen deficiency. Subchronic administration of α7nAChRs antagonist methyllycaconitine to rats with bilateral chronic ligation of the common carotid arteries reduces the delayed mortality in animals by half or more (data unpublished).

Thereby, we have reason to assume the existence of pathway of filtering (inhibition) of cognitive stimuli in the hippocampus via α7nAChRs under HBH conditions.

### 3.4. Probable Mechanism of the Potentiation of Hypoxic Preconditioning by DMSO: A Potential Intersection Point with PNU and HBH

iHippo administration of 0.05% DMSO, as well as IP administration at a concentration in the brain of under 0.04%, potentiated the efficiency of the preconditioning effect of HBH in rats with PPI ≥ 40% by 1.8–2.0-fold. PNU dose-dependently negated this action of DMSO.

This suggests that DMSO does not act as an anticholinesterase agent on α7nAChRs. This is possible as α7nAChRs are low affinity receptors, unlike other nAChRs [38], and high doses of acetylcholine are required to activate them [31,38,39]. However, at the concentrations used, DMSO could decrease the activity of cholinesterase by only a few percent [26,27]. If α7nAChRs are not activated under conditions of cholinergic activation of HBH, then a small influx of acetylcholine initiated by DMSO was also unable to act on this receptor subtype. 

In recent years, research has been published on the effect of DMSO at low concentrations, opposing the procognitive effect of α7nAChRs agonists [40]. The authors found that DMSO at a concentration of 0.05% in the CA1 area of the hippocampus significantly affects the intrinsic excitability of CA1 pyramidal neurons. It reduces their excitability, reducing the frequency of miniature excitatory postsynaptic potentials and, thereby, the action potential output for any given stimulus. Thus, DMSO induces hypoexcitability of neurons that play a key role in long-term potentiation (LTP) mechanisms.

The activation of α7nAChRs in the CA1 area is an integral component of the GABAergic network, which initiates the synchronization of glutamatergic pyramidal neurons (LTP) in cognitive processes. At the electrophysiological level, activation of α7nAChRs enhances LTP in rat septo-hippocampal slices and increases GABAergic inhibitory postsynaptic currents (IPSCs), which modestly hyperpolarize pyramidal neurons [31]. As we noted above, this is a classic electrophysiological response in the CA1 neuronal procognitive network of the hippocampus. However, it is important that this response of pyramidal neurons is recorded in the first seconds to hundreds of milliseconds of stimulation [3,31,33].

A recent study traced the temporal dynamics of plasticity of pyramidal neurons and GABAergic interneurons in the CA1 procognitive network over 60 min [41] (the usual registration time for LTP or theta oscillation [8,31,33,41]). Using a hippocampus-dependent inhibitory avoidance task, the authors analyzed the dynamic changes in learning-induced synaptic plasticity on the acute brain slices which were taken from the rat brain at different stages after a one-time training (a scrambled electrical foot-shock). Step by step control revealed that: (1) In pyramidal neurons, a decrease in the frequency of miniature excitatory postsynaptic potentials occurred only in the first minute after training, after five minutes the frequency increased sharply, and by 60 min it was approaching normalization; (2) the initiators of plastic changes in the network were GABAergic interneurons, in which the frequency of miniature inhibitory postsynaptic potentials increased immediately, in the first minute after training, and remained elevated throughout the experimental period; and (3) throughout this period, the amplitude of miniature potentials, exciting and inhibitory, was increased. In other words, the LTP period is characterized by an increased functional state of not only inhibitory interneurons, but also pyramidal neurons.

In the study [40], the authors emphasized that the suppressive effect of 0.05% DMSO on the excitability of pyramidal neurons was not the result of acute exposure to the solvent, but after 2 to 5 h of incubation of hippocampal slices. This time period is consistent with our SHBH T test time (within 1.5–2 h of drug administration). Thus, it is reasonable to assume that 0.05% DMSO has an inhibitory effect on pyramidal neurons under conditions of hypoxic preconditioning, at least in CA1 neurons. This suggests the participation of the solvent in the mechanism of filtering cognitive stimuli under HBH conditions. This ability of DMSO alone can provide a potentiating effect on HBH efficiency in rats with PPI ≥ 40%. This could explain the suppressive effect of PNU in our experiments on the effect of DMSO, as a strong selective agonist of receptors that trigger the GABAergic procognitive network.

The targets of this action of 0.05% DMSO on CA1 pyramidal neurons are unknown [40]. The authors, due to the specifics of the action of DMSO, suggest that the target may be Cl^−^ channels. It has been shown in vitro that DMSO at high concentrations (0.3–3%) dose-dependently inhibits GABA-induced Cl^−^ currents [42]. Considering the anticholinesterase properties of the solvent, we cannot exclude that DMSO can act in a complex or only through cholinergic receptors. For example, GABAergic interneurons in the procognitive network of the hippocampus, in addition to α7nAChRs, express the nAChR α4b2 subtype, potentially at a higher density [28,31]. α4b2 nAChRs are represented by both low and high affinity receptors, and in the second case, the effective concentrations of acetylcholine are several orders of magnitude lower [38,43]. It is also important that the neural network in CA1 of the hippocampus is designed in such a way that it includes several types of GABAergic interneurons that differ in the nature of connections with pyramidal neurons and each other. The activation of nAChRs can induce either inhibition or disinhibition of interneurons, and their effect on the efferent pyramidal neurons depends on their connections in the GABA-glutamatergic network [28,31,44,45].

The relationship between the mechanisms of action of DMSO and HBH is clear. The study found a stable negative correlation between PPI-associated DMSO and HBH values in rats with PPI ≥ 40%. The lower the efficiency of HBH itself, the more effective DMSO was at influencing it. The level of correlation was highest in the DMSO iHippo group, further highlighting the key role of the hippocampus in the HBH mechanism. The negative nature of the DMSO-HBH relationship indicates that the targets of the action of DMSO and hypoxic stimuli are different. This further reinforces the idea that DMSO participates not in the preconditioning mechanism of HBH, but, most likely, in the filtration mechanism, and allows us to put this forward as a working hypothesis.

The PPI-associated relationship between these two mechanisms is clear and, given the iHippo nature of both, strengthens our conclusion that the hippocampus acts as a physiological substrate of cross-correlation relationships between different patterns of adaptive-compensatory mechanisms [25].

### 3.5. Are α7nAChRs Not Involved in Hypoxic Preconditioning?

Previous data in the literature has suggested that α7nAChRs play a positive role in the mechanisms of preconditioning and anti-ischaemic protection (see Introduction). However, we believe this is due to the individual’s response to the drug.

In this study, we reported the results of rats with PPI ≥ 40%. At the same time, in the subgroup of rats with PPI < 40%, PNU after IP administration potentiated the efficiency of HBH. The α7nAChRs showed a positive effect on the hypoxic preconditioning mechanism [23,25]. HBH in this subgroup of rats implements a preconditioning effect according to some other functional scheme than in rats with PPI ≥ 40%, through inhibition of synaptic activity in cholinergic projections. However, the physiological meaning of the positive effect of α7nAChRs on hypoxic preconditioning remains to be elucidated.

## 4. Materials and Methods

### 4.1. Animals and Ethical Approval

The pharmacological experiments were carried out in male outbred albino laboratory rats aged 3–4 months (weight 300–450 g). The animals came from the Light Mountains nursery (Moscow, Russia) and were kept in the vivarium of the Institute of General Pathology and Pathophysiology. All animal care and experimental procedures were carried out in accordance with the European Communities Council Directive of 24 November 1986 (86/609/EEC). The suffering of experimental animals and their number has been minimized. All experimental protocols were approved by the Ethical Committee of the Institute of General Pathology and Pathophysiology (protocol #3 of 18.08.2021). The rats were housed in a temperature-controlled room (20–24 °C) and were maintained with a 12 h light-dark cycle. They were kept at 5–7 animals per rat cage of 40 × 60 cm in size, and had free access to food and water. 

After the pharmacological experiments, the rats who underwent iHippo injection were euthanized by inhalation of carbon dioxide in the system for animal euthanasia (Open Science, Krasnogorsk, Russia), then the brain was removed and used as a morphological control. The rats who underwent IP administration were euthanized using the same euthanasia apparatus.

### 4.2. Acoustic Startle Reaction Model

The acoustic sensorimotor startle reaction model (PPI Test) was measured using the TSE startle response system (TSE System, Bad Homburg, Germany). The rats were tested in soundproof boxes within restraining metal cages installed on the platform (27 × 9 × 19 cm). The acoustic stimuli were presented from two loudspeakers, which were mounted along the walls of the metal cages. Broadband noise with duration of 100 ms and loudness of 110 dB was used as the main stimulus. The background-masking sound signal had a broadband noise with a loudness of 72 dB, and the prestimulus (prepulse) had signal duration of 40 ms and loudness of 85 dB. The animals were exposed to a total of 12 trials after 5 min adaption. The first two trials were pulse-alone trials (habituation). The remaining 10 trials were presented in pseudo-random order and included five pulse-alone trials and five pulse trials with a preceding prepulse of 100 ms in the lead-off interval. The inter-trial intervals had a mean value of 15 s (from 10 to 20 s). The PPI value was estimated using the formula (Am − Ap)/Am × 100%, where Am is an average reaction amplitude in the samples without the pre-pulse, excluding the first two (n = 5), and Ap is an average reaction amplitude in the samples with the pre-pulse (n = 5) [25].

### 4.3. Surgical Procedures

The implantation of guiding cannulas was performed 2 days before intracerebral administration of drugs. All animals were anaesthetized by intraperitoneal injection of zoletil at a dose of 15 mg/kg and xelazine at a dose of 2 mg/kg. Then the rats were scalped and fixed in Stereotaxis (TSE Systems, Bad Homburg, Germany). Holes were drilled in the skull along the coordinates (AP − 3.6; DL ± 2.0) [46]. Stainless steel guiding cannulas with a height limiter (H − 2.0) were introduced into the holes. The cannulas were fixed to the skull using dental cement, and the stainless steel mandrels were placed in the hollow hole of the cannula. Then the animal was returned to its home cage, where for the next 2 days it was given free access to food and drink.

### 4.4. Drug Administration

#### 4.4.1. IP Drug Administration

Rats received a single IP injection of 3% DMSO, the solvent for PNU, in saline (DMSO IP group) in a volume of 1 mL/kg of rat weight or, similarly, a single IP injection of saline (Saline IP group).

#### 4.4.2. iHippo Drug Administration

All drugs were injected bilaterally at a depth of 2.7 mm (CA1 area of hippocampus) using a 10 μL syringe (Hamilton- # 701, Reno, NV, USA) in a volume of 1.5 μL for 1 min. In each hippocampus, PNU was injected at 0.7075 μg (0.00262766945 μmol) in 3% DMSO diluted in saline (PNU iHippo group). DMSO in saline was administered at 3% (DMSO iHippo group) or saline only (saline iHippo group). The rat brain weighs approximately 2 g and has a volume of 3.5–4 mL. Each hippocampus weighs approximately 50 mg and its extreme values are 40 and 60 mg. Based on these data, each hippocampus occupies a volume of approximately 87.5 μL and 120 and 70 μL at its extreme values, and then the PNU concentration in the hippocampus was approximately 30 μM, with the extreme values 37.5 and 22 μM, respectively. Using a similar calculation, the final concentration of DMSO in the hippocampus was 0.05%, with extremes of 0.06 and 0.04%, respectively.

### 4.5. Hypoxic Models

Both hypoxic models were described in detail [23,25]. Varying severities of hypoxia were induced in different pressure chambers. One chamber was for the simultaneous testing of two or more rats (branded Pressure chamber from the manufacturer “Schrödeu.Co.”, Lübeck, Germany). Other pressure chambers, for individual rat testing, were glass transparent containers of 3 L volume, with a vacuum gasket and cover equipped with a needle valve and valves for a vacuum pump and altitude gauge inlets that were custom fabricated. Each of them was connected to a single stage vacuum pump (model VP 115, 2016, Guangdong, China) and to an altitude gauge, which was calibrated to an altitude above sea level (model VD-20, # 152300254, 2016, Zaporizhzhia, Ukraine). The rats in the chamber were “raised” at a speed of 12–15 m/s to the adaptive altitude of 5000 m (HBH, 3.0 Pa, equivalent to 10–11% O_2_, 60 min) and at a speed of 63.5–64 m/s to the critical altitude of 11,500 m (SHBH, 1.2 Pa, equivalent to 4.5% O_2_). The resistance to hypoxia at the critical altitude test was recorded as the time (T) until agonal inspiration (apnoea) in combination with a loss of control of body tone.

### 4.6. Postmortem Brain Analysis

After the experimental test, the rats were euthanized in the system for animal euthanasia (see Section 4.1). Then, using a guillotine, decapitation was performed, followed by extraction of the brain for histological analysis. Frontal brain sections were made on a Microm HM 505E cryostat (Micron International, Walldorf, Germany). They were fixed on a glass slide. The next day, we visually assessed the correct position of the guide cannulas in the brain using a microscope (Micromed MC-5ZOOM LED, Petersburg, Russia).

### 4.7. Experimental Protocol 

Each rat was handled for at least two consecutive days before starting the experimental procedures. 

The rats were tested using the PPI Test and the values of PPI were estimated. For further experiments, rats with PPI ≥ 40% (n = 33) were selected. According to their PPI values, the rats were subdivided into the control groups, saline IP (n = 8), saline iHippo (n = 5), and PNU iHippo (n = 7) group, and the DMSO groups, the DMSO IP group (n = 6) and DMSO iHippo group (n = 8). Two or three days later, pharmacological experimental procedures began. 

The test intervals and IP time of drug administration before HBH training have been verified previously [23,25]. In the rats from the IP groups, the drugs (DMSO in saline or saline alone) were administered 20–25 min before the start of HBH training and 4 min after its end were also exposed to SHBH and the values of T were estimated. 

For iHippo injections, animals were cannulated into the hippocampus in small batches in order to undergo subsequent HBH and SHBH tests on the third day after surgery. The mandrins were removed from the cannulas in the rat just before injection. All rats in the iHippo groups received PNU in DMSO and saline, DMSO in saline, or saline injections into the left and right hippocampus in a random sequence. Immediately after injection into the second hippocampus (total injection time was under 10 min), the rats were subjected to the HBH session, and 4 min after its end they were exposed to SHBH and the values of T were estimated.

The data were obtained in a blinded manner. Each experimenter did not know the key characteristics of the tested rat at the stage of obtaining experimental data, because the preparation of the animals for the experiment was carried out by another experimenter. In addition to this, the experimental animals were passed into a new room and environment after each test. 

Remarks: During the removal of the mandrins, fixed dental cement fell off along with cannulas in one rat from the saline iHippo group. We tested this rat without saline as a control for surgical procedures (pure control). Another rat recruited to the PNU iHippo group was susceptible to anesthesia and died after surgery prior to pharmacological procedures.

### 4.8. Drugs

The following drugs were used: PNU-282987, Tocris Bioscience (Bristol, UK); DMSO, LLC Tula Pharmaceutical Factory (Tula, Russia); saline, REACHIM (Old Kupavna, Moscow Region, Russia). 

### 4.9. Data Analysis

Before starting the analysis of the results, the data normality was assessed with STATISTICA 8.0 using the Kolmogorov–Smirnov test (parameter d and *p*-values). No deviations from normality were found. 

The data were then statistically tested using the non-parametric Wilcoxon–Mann–Whitney test (u-criterion) and Fisher’s exact test (FET criterion). We also performed a correlation analysis using Pearson’s r-criterion in Microsoft Excel, using the correcting formula for small samples n ≤ 15 [47]. The T values of compared variation series in our pharmacological groups included a similar range to the PPI values. 

The differences were considered significant at *p* < 0.05. The Holm-Bonferroni method was used to identify significant differences when comparing the data from the saline (Control), PNU, and DMSO groups. 

The T value, as a function of PPI, increases linearly in proportion to the decrease in PPI value. Therefore, the results were presented as a percentage of the ideal T value, which was calculated using the linear trend formula built by Excel on the basis of the entire PPI-T test array in the group of control rats that received saline injections before HBH (n = 46). Since we are comparing our data sets, highlighting the subgroup with PPI > 40%, the comparison of all data as a percentage of the true control value made this comparison valid.

In both saline groups, iHippo and IP, two similar T values were calculated as the “outliers”. In each of these groups, T values with similar PPI values were more than two times higher than the average value of variation series and was excluded from the analysis.

## 5. Conclusions

We have previously identified that in rats with PPI > 40%, HBH in vivo activates the cholinergic projections of hippocampus and the IP administration of DMSO (solvent of PNU and the anticholinesterase agent at the concentrations used) potentiates the preconditioning action of HBH, while PNU (the selective α7nAChRs agonist) suppresses the effects of HBH and DMSO [25].

Now we carried out an in vivo pharmacological study of the effect of PNU and DMSO on the efficiency of preconditioning action of HBH when both drugs administered to the CA1 hippocampus area in rats with PPI ≥ 40%. 

The results of iHippo injections for both PNU and DMSO were in good agreement with the effects of both drugs when administered IP. Thus, direct evidence has been obtained that the mechanisms of hypoxic preconditioning are formed in the hippocampus.

Moreover, experiments with iHippo administration made it possible to clarify the nature of the relationship between PNU and DMSO with hypoxic preconditioning and with each other.

The suppressive effects of PNU on HBH efficiency are realized through the activation of α7nAChRs at all agonist concentrations used and for all routes of administration. Therefore, we conclude that the activation of α7nAChRs do not participate in cholinergic activation initiated by HBH in rats with PPI ≥ 40%. The activation of α7nAChRs is recognized as a mandatory link in the procognitive network of the hippocampus. It is reasonable to assume that inhibition of α7nAChRs through the mechanism of filtration of cognitive signals may enhance hypoxic preconditioning.

The opposite direction of the PNU and DMSO effects indicates that the activation of α7nAChRs is also not involved in the anticholinesterase effect of DMSO.

The efficiency of positive effect of DMSO on HBH was inversely correlated with the efficiency of preconditioning effect of HBH itself. Therefore, a working hypothesis has been put forward that DMSO at a concentration of 0.05% or less (with IP administration) potentiates the efficiency of HBH hypoxic preconditioning in rats with PPI ≥ 40% indirectly, for example, through the filtering of extraneous signals. It is possible that the mechanisms of filtering are the point of intersection of the effects of DMSO and PNU under HBH conditions. 

The assumptions that follow from the results of this study can be verified experimentally, and this is included in the plans of our further experimental work.

The neural mechanisms of nervous functions are being studied increasingly actively. At present, scientific efforts are actually directed only at the neuronal organization of cognitive functions and the hippocampus and its α7nAChRs are rightfully one of the main targets of these studies. Our study is the first direct evidence that the hippocampus is a key structure in the formation of hypoxic preconditioning, as well as that activation of α7nAChRs is not involved in the cholinergic mechanism of hypoxic preconditioning in rats with PPI ≥ 40%.

Knowledge of the neuronal mechanisms of hypoxic preconditioning is of fundamental importance in understanding the organization of brain functions and their interactions, and also represents an important therapeutic potential for clinical use.

## Figures and Tables

**Figure 1 molecules-26-07387-f001:**
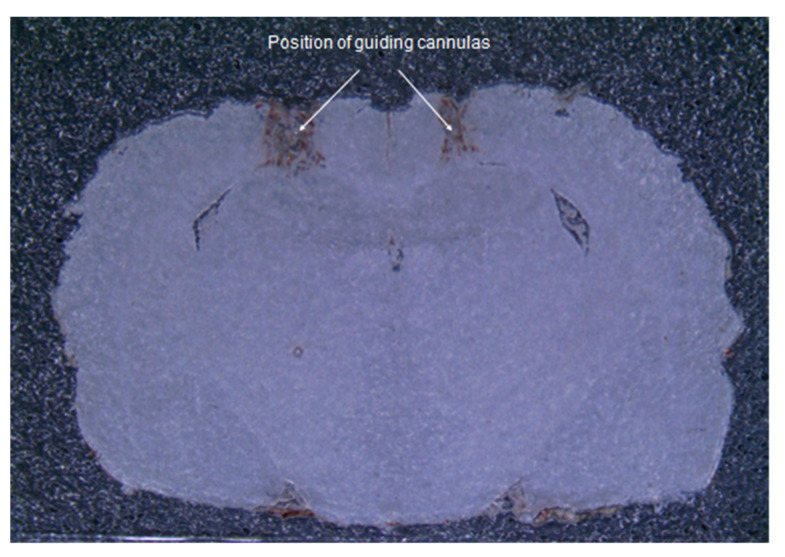
Frontal section of the brain. A photograph of a brain slice of an operated rat demonstrates the correspondence of insertion sites of guide cannulas to the selected coordinates.

**Figure 2 molecules-26-07387-f002:**
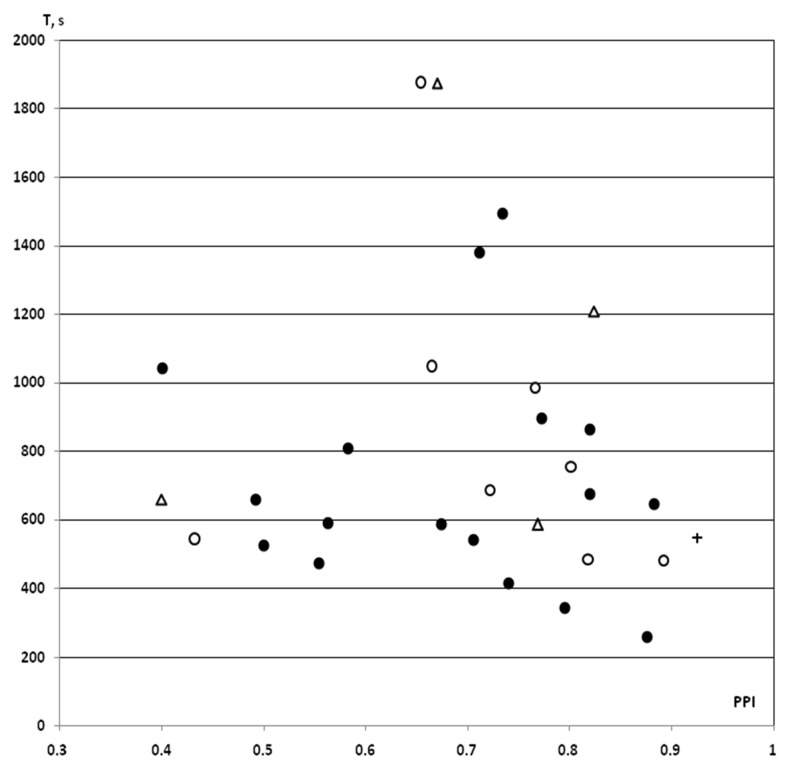
PPI-associated individual values of resistance to hypoxia in control rats with PPI ≥ 40% exposed to SHBH 4 min after the end of the HBH session. HBH, moderate hypobaric hypoxia (hypoxic preconditioning); SHBH, severe hypobaric hypoxia. T (s, ordinate axis), a time until agonal respiration (apnea) under SHBH conditions; PPI (abscissa axis), prepulse inhibition in the acoustic sensorimotor startle reaction model. The PPI values are presented in decimal digits. On the graph, individual values of T corresponding to own PPI values. Dark round markers, the T values of saline IP group obtained in previous experiments (saline + HBH, IP introduction, n = 17); light circles markers, the T values of saline IP group in this study (saline + HBH, IP introduction, n = 8); light triangles markers, the T values of saline iHippo group in this study (saline + HBH, iHippo introduction, n = 4); cross marker, control for surgical procedures, the T value of this study obtained on a rat implanted with cannulas (HBH only, without saline introduction). In the saline IP and iHippo groups, the one value of T in each of them calculated as “outliers” was excluded from the analysis (both T > 1800 s). The T values on the graph represented a same variation series (see also Figure 3). Therefore, the data from this study were combined into a control group with the exception of “outliers” (n = 11, A in Figure 4 and Figure 5).

**Figure 3 molecules-26-07387-f003:**
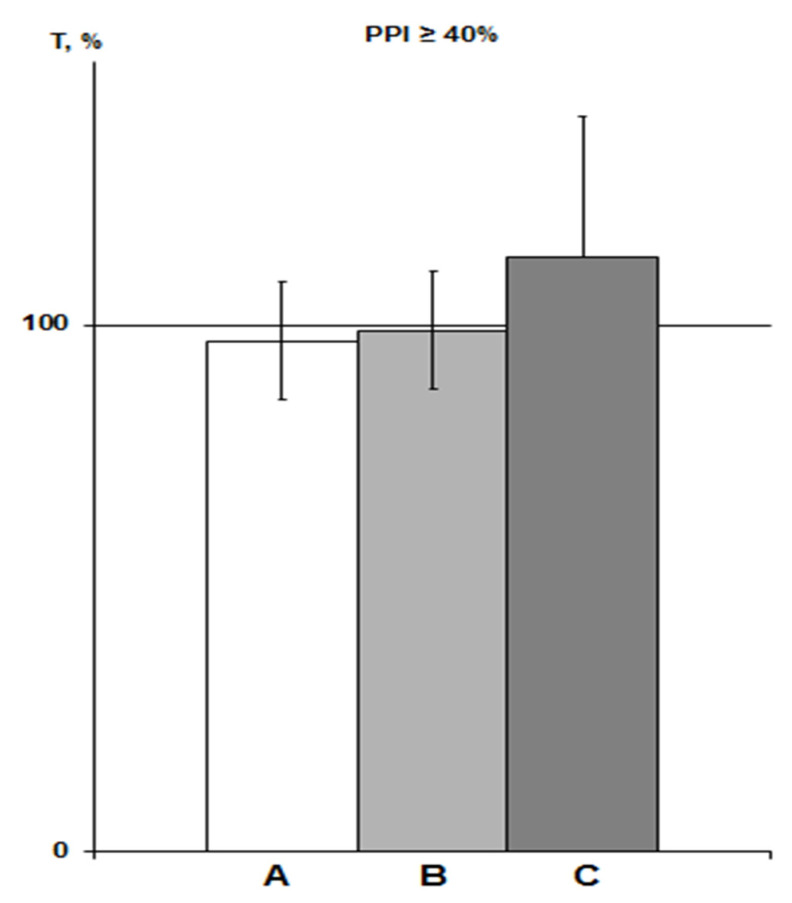
PPI-associated average T values in various saline groups of rats with PPI ≥ 40% exposed to SHBH 4 min after HBH. The T values (means ± SEM%) are expressed as a percentage of ideal T values as a function of the corresponding PPI (see Section 4.9). A, saline IP group of previous experiments (n = 17); B, saline IP group of this experiment (n = 7); C, saline iHippo group of this experiment (n = 3).

**Figure 4 molecules-26-07387-f004:**
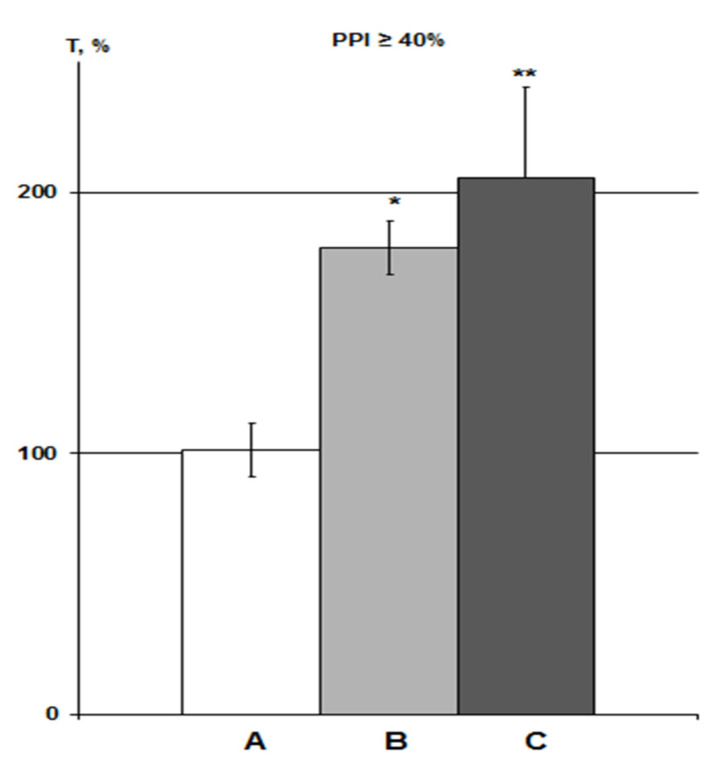
Influence of DMSO on HBH efficiency with IP and iHippo administration. The T values are expressed as in Figure 3 (means ± SEM%). A, control group (n = 11); B, DMSO IP group (n = 6); C, DMSO iHippo group (n = 8). * significant differences with the control group. * *p* < 0.05; ** *p* < 0.02. The Wilcoxon–Mann–Whitney test (u-criterion) and Fisher’s exact test (FET-criterion). *p* values are presented as amended by the Holm-Bonferroni method.

**Figure 5 molecules-26-07387-f005:**
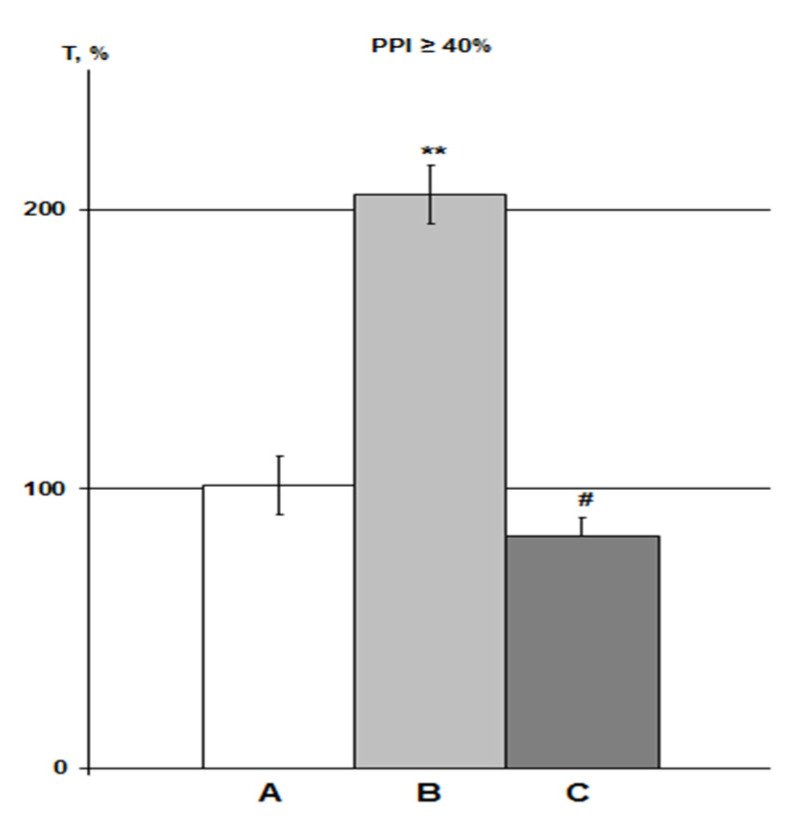
Effect of PNU on DMSO action on HBH efficiency with iHippo administration. The T values are expressed as in Figure 3 (means ± SEM%). A, control group (n = 11); B, DMSO iHippo group (n = 8) as ‘C’ in Figure 4; C, PNU iHippo group (n = 6). On the graph, PNU at a concentration of 30 μM completely cancels the potentiating effect of DMSO on HBH efficiency and reduces the preconditioning effect of HBH by 18%, but this decrease is not significant. * significant differences with the control group; # significant differences with the DMSO iHippo group. ** *p* < 0.02; # *p* < 0.03. The Wilcoxon–Mann–Whitney test (u-criterion) and Fisher’s exact test (FET-criterion). *p* values are presented as amended by the Holm–Bonferroni method.

**Table 1 molecules-26-07387-t001:** Correlation analysis between the variational series of DMSO and control groups. Correlation between the T values of control group with the combined DMSO group (NN 1–2), DMSO IP group (NN 1–3), and DMSO iHippo group (NN 1–4). See Section 2.2.2 for details. Pearson’s r-criterion. *p* values are presented as amended by the Holm-Bonferroni method.

NN	Injection tipe	IP	iHippo	iHippo	IP	iHippo	iHippo	IP	iHippo	iHippo	IP	IP	iHippo	IP	iHippo	n	r-Criterion	P with Amended Holm-Bonferroni
1	PPI%	40.6	49.2	56.3	66.5	67.4	70.5	72.2	74	76.9	79.5	81.9	87.6	88.2	89.2	14		
	T, % Control	121	60	75	141	79	74	96	58	83	50	127	39	98	74			
																	NN 1–2	
2	PPI%	43	53.4	54.3	62.5	68.8	69.8	70.5	74.9	75.5	78	82.2	86	87.6	89.1	14		
	T, % DMSO, iHippo+IP	105	158	194	136	226	89	175	335	130	352	159	389	148	124		−0.669	<0.03
																	NN 1–3	
3	T, % DMSO, IP	105			136			75			352	159		148		6	−0.912	<0.04
																	NN 1–4	
4	T, % DMSO, iHippo		158	194		226	89		335	130			389		124	8	−0.825	<0.02

## Data Availability

The data presented in this study are available from the authors.

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
