# Peer review of "Effect of Intrahippocampal Administration of α7 Subtype Nicotinic Receptor Agonist PNU-282987 and Its Solvent Dimethyl Sulfoxide on the Efficiency of Hypoxic Preconditioning in Rats"

_molecules, 2021, doi:10.3390/molecules26237387_

Round 1
Reviewer 1 Report
The manuscript was improved, and most of the comments were addressed by the authors. Some minimal comments are added here:
Results:
Page 6, first paragraph: the percentage of the parameter T (the HBH efficiency) increased by 79% in the DMSO IP group, whereas the difference between DMSO IP group and DMSO iHippo group was 18%. According with the height of the column C, which is higher than 200% (Fig. 4), this difference is more than 18%. Please, check the values.
From the previous review:
Comment (Discussion):
Although the information contained in this section is interesting, some of this seems to be not close related with the main theme of the work; for instance, page 9, last paragraph. Further, some parts are very speculative.
Our answer:
We have removed this and the previous paragraphs, as well as several fragments from the proposals in the Discussion, which are not directly related to the main topic of the work.
New comment:
These changes made for the authors do not correspond to those mentioned in this comment; “… for instance, page 9, last paragraph”, which is referred to the first version of the manuscript.
Reviewer 2 Report
When the main conclusions in Elena I Zakharova et. al., Brain Sci., 2020, is that “hippocampal cholinergic mechanisms are involved in hypoxic preconditioning via nicotinic α7 receptors”, I am not sure on what basis authors are claiming that they are establishing it for the first time the role of the hippocampal a7 in hypoxic preconditioning? Unless I am missing something obvious.
Having said, intrahippocampal administration of PNU was done for the first time that adds value to the study and certainly confirms the role of hippocampus in hypoxic preconditioning via nicotinic α7 receptors. This study is a logical extension of their previous study published in Brain Sci. I agree to the fact that perhaps the most important takeaway from the study is to establish and confirm that cholinergic mechanism of hypoxic preconditioning is not via a7 activation. Authors should certainly highlight this given the amount of attention a7 receptors have on cholinergic signaling. In fact authors might want to consider this as the basis of doing this study - to ascertain if a7 activation was necessary in hypoxic preconditioning? This according to me would be a stronger basis than exploring the role of a7 desensitization as the reason for direct intrahippocampal administration. I leave it, however, to the discretion of the authors.
The quality of English writing was significantly improved to the standards of the publication. Possible mechanisms explaining the effect when DMSO was administered was informative.
Minor comments: Check references to maintain a consistent font style, size.
Author Response
Please see the attachment.

This manuscript is a resubmission of an earlier submission. The following is a list of the peer review reports and author responses from that submission.
Round 1
Reviewer 1 Report
The present manuscript has been correctly focused on the topic.
The paper is well organized in each section.
Methods are clearly described as the findings.
Figures and Tables are explicative and useful.
English is correct.
No changes are required.
Reviewer 2 Report
The present manuscript by Zakharova co-workers is related with the effects of both PNU-282987 (agonist of the α7 nicotinic receptor, nAChRs) and DMSO (dimethyl sulfoxide) on the efficiency of hypoxic preconditioning in rats.
The main findings were that in rats with acoustic startle prepulse inhibition (PPI) > 40%, a) DMSO potentiates the preconditioning efficiency of hypobaric hypoxia (HBH) administered intraperitoneally as well as with intrahippocampal injections, and b) PNU-282987 reduces HBH efficiency with intrahippocampal injections.
The subject of this study is relevant and interesting. The manuscript is clear, and the experiments are well designed and performed.
Comments for the manuscript:
Abstract:
It is not clear for this reviewer why the authors hypothesized that PNU-282987 desensitizes α7 nAChRs; taking in consideration the kinetic properties of this receptor, see also reference 3, in which the responses mediated by α7 nAChRs desensitize.
Regarding the sentence: “PNU-282987 acts as agonist via any route of administration”, please argument if another type of response or property would be expected with this substance.
Introduction:
First paragraph, list the references as [2-4, 6-9], [2-7].
Page 2, second paragraph: “PNU, in contrast, reduced the effects of DMSO on HBH …”. How do the authors would know the actions of PNU-282987 on the effects of DMSO, if the agonist is dissolved in DMSO? How do they can isolate the effects of PNU-282987 from DMSO?
Results:
Page 5, second paragraph: the percentage of the parameter T (the HBH efficiency)
increased by 179% in the DMSO IP group, whereas the difference between DMSO IP group and DMSO iHippo group was 18%. According with the height of the column C (Fig. 4), this difference is higher. Please, check the values.
Discussion:
Although the information contained in this section is interesting, some of this seems to be not close related with the main theme of the work; for instance, page 9, last paragraph. Further, some parts are very speculative.
Conclusions:
This section has content that may move to the Discussion section.
In this way, it would be beneficial for the reader if the conclusions are presented in a shorter text, focusing on conclusions directly obtained from the results.
Page 14, first paragraph: “Therefore, we conclude that α7nAChRs do not participate in cholinergic activation initiated by HBH in rats with PPI > 40%”. In here, which would be the candidate for cholinergic activation?
Reviewer 3 Report
The current work is an extension of the work done by the same group where they showed the role of cholinergic neurons in the hippocampus, and the action of the a7 selective agonist PNU-282987 and the solvent dimethyl sulfoxide (DMSO). Here they show that, like the intraperitoneal administration, intrahippocampal injections of PNU-282987 into the hippocampal area decreased the HBH efficiency in rats with PPI ≥ 40%. The role of a7 in the hippocampus in pre-conditioning effect of the HBH has previously been established.
The key addition of the current study, over the previous study that the same group published before, is injecting the a7 selective agonist PNU-282987 directly into the hippocampus (CA1 area). As mentioned, the authors found no difference in the HBH efficiency when PNU-282987 was administered intraperitoneally or injected directly into the hippocampus (iHippo). The reasoning given by the authors to test hippocampal injection of the PNU-282987 is that the a7 receptors could have been desensitized when administered via IP. I find this reasoning very poorly conceived for a couple of reasons. Firstly, it’s a well-known fact that nicotinic receptors, especially the a7 subtype, are fast desensitizing receptors that transition into closed desensitized state in the order of milli seconds that follow receptor opening; and unless you are measuring the response close to that time scale, you can never account for receptor desensitization. Secondly, the DMSO is known to inhibit several nicotinic acetylcholine receptors, and if DMSO potentiates the effect, an agonist should the opposite, which is inhibition of DMSO induced potentiation, as observed. Hence, there was no reason to suspect the role of receptor desensitization. More than the negative result (no change in the HBH efficacy by direct injection), the problem lies in the hypothesis and the manner the experiments were conceived. One justification for direct injection into the hippocampus could have been the ability of the PNU-282987 to cross the blood brain barrier (BBB), but it is already established that the small molecule in the study crosses the BBB, which includes studys done by the same group previously. In addition to the fact that we are not learning anything new, the manuscript is poorly written with errors in sentence construction and grammar.